# Characterize direct protein interactions with enrichable, cleavable and latent bioreactive unnatural amino acids

Dan-Dan Liu[1,2,8], Wenlong Ding [1,2,8], Jin-Tao Cheng [1,2,8], Qiushi Wei [3,8], Yinuo Lin[4], Tian-Yi Zhu[1,2], Jing Tian[4], Ke Sun[5], Long Zhang [1], Peilong Lu [5], Fan Yang [6], Chao Liu [3] ✉, Shibing Tang [4,7] ✉ & Bing Yang [1,2] ✉

Latent bioreactive unnatural amino acids (Uaas) have been widely used in the development of covalent drugs and identification of protein interactors, such as proteins, DNA, RNA and carbohydrates. However, it is challenging to perform high-throughput identification of Uaa cross-linking products due to the complexities of protein samples and the data analysis processes. Enrichable Uaas can effectively reduce the complexities of protein samples and simplify data analysis, but few cross-linked peptides were identified from mammalian cell samples with these Uaas. Here we develop an enrichable and multiple amino acids reactive Uaa, eFSY, and demonstrate that eFSY is MS cleavable when eFSY-Lys and eFSY-His are the cross-linking products. An identification software, AixUaa is developed to decipher eFSY mass cleavable data. We systematically identify direct interactomes of Thioredoxin 1 (Trx1) and Selenoprotein M (SELM) with eFSY and AixUaa.

Identifying binding partners of target proteins is an effective approach to reveal functions of target proteins, and the construction of protein-protein interaction (PPI) networks is essential to the understanding of biological processes at the system level[1,2]. Affinity Purification Mass Spectrometry (AP-MS) is a key method for PPI study by isolating and identifying interacting proteins from cell lysates[3]. However, lysing cells can disrupt structures of subcellular organelles and dilute protein concentrations, thus, perturbing physiological protein binding and causing artificial protein interactions[4]. Genetically encoded latent bioreactive unnatural amino acids (Uaas) enable capturing protein interactions in live cells[5–7], and protein direct interactions and their interaction interfaces can be mapped by the identification of Uaa mediated cross-linked peptides[8,9]. Meanwhile,

the cross-linked peptides can improve accuracy of protein structure prediction[10].

Dozens of latent bioreactive Uaas have been developed to add covalent bonds to proteins[11,12]. However, it is still challenging to identify Uaa-mediated cross-linked peptides due to the complexity of protein samples and the low ionization efficiency of cross-linked peptides. Peptide level enrichment has been demonstrated to significantly increase the detection sensitivity of cross-linked peptides obtained from small molecular chemical cross-linkers[13,14]. A series of enrichable photo-crosslinking Uaas have been designed and applied to capture protein interactions, but few cross-linked peptides were identified because of their low cross-linking efficiency[15]. An enrichable latent bioreactive Uaa EB3 has been successfully applied on model

[1]Life Sciences Institute, Department of Medical Oncology, The Second Affiliated Hospital of Zhejiang University School of Medicine, Zhejiang University, Hangzhou, Zhejiang 310058, China. [2]Cancer Center, Zhejiang University, Hangzhou, Zhejiang 310058, China. [3]School of Biological Science and Medical Engineering & School of Engineering Medicine, Beihang University, Beijing 100191, China. [4]State Key Laboratory of Respiratory Disease, Center for Chemical Biology and Drug Discovery, Guangzhou Institutes of Biomedicine and Health, Chinese Academy of Sciences, Guangzhou, Guangdong 510530, China. [5]Key Laboratory of Structural Biology of Zhejiang Province, School of Life Sciences, Westlake University, Hangzhou, Zhejiang 310030, China. [6]Department of Biophysics, Kidney Disease Center of the First Affiliated Hospital, Zhejiang University School of Medicine, Hangzhou, Zhejiang 310058, China. [7]China-New Zealand Joint Laboratory on Biomedicine and Health, Guangzhou 510530, China. [8]These authors contributed equally: Dan-Dan Liu, Wenlong Ding, Jin-Tao Cheng, Qiushi Wei. ✉e-mail: liuchaobuaa@buaa.edu.cn; tang_shibing@gibh.ac.cn; bingyang@zju.edu.cn

proteins in live cells[8], but it mainly targeted cysteine and abundance of cysteine in the whole proteome is low. Therefore, an Uaa that is multiple amino acids reactive, highly efficient at protein cross-linking, and enrichable is urgently needed.

Fluorosulfate-L-tyrosine (FSY) is a nontoxic Uaa with reactivities to lysine, histidine and tyrosine residues[16]. The cross-linking reaction of FSY has been demonstrated to be high efficient both in vitro and in live cells. In addition, FSY has been applied to developing covalent protein drugs, nanobody[17–21], and capturing protein-RNA interactions[22]. However, off-target effect of FSY incorporated protein binders has been evaluated with SDS-PAGE gel and western blot but not MS[23], and FSY mediated protein-RNA cross-linking products have not been validated by MS. Recently, A series of FSY analogs have been developed to capture protein binders[21,24–26], but none of them are enrichable Uaas. Therefore, it is necessary to design an enrichable Uaa based on FSY to enhance the detection of FSY-induced cross-links in order to comprehensively map FSY-captured targets.

MS cleavable small molecular cross-linkers have been widely used to study protein structures and protein-protein interactions[27–30], and chemically cleavable photo-crosslinking Uaa has been developed and applied to characterize protein-protein interactions[31,32]. However, till now, there is no report of MS cleavable and chemical cross-linking Uaa (MSCXUaa). We believe this kind of Uaa has multiple advantages. First, in comparison with small molecular cross-linkers, it can be applied to live cells without killing cells, since protein cross-linking is generated by MSCXUaa in situ and in physiological conditions. Second, compared with chemically cleavable photo-crosslinking Uaa, MSCXUaa will keep cross-linked peptides integrated in MS1 analysis, which will increase the identification specificity of cross-linked peptides. Lastly, because the ionization efficiency of non-cleavable cross-linked peptides is low[33] and it is not easy to obtain large fragment ions across the cross-linking site, relative low fragment ions coverage of cross-linked peptides is typically observed. If the cross-linking bond can be fragmented in MS, this will enhance fragment ions coverage of cross-linked peptides[34]. Therefore, we believe that a MSCXUaa and software for deciphering MS data of MSCXUaa will be helpful for mapping Uaa mediated cross-linking.

Here, we report a latent bioreactive Uaa eFSY that is multiple amino acids reactive and enrichable, and demonstrated that eFSY mediated Uaa-Lys and Uaa-His cross-links are MS cleavable. To make use of the MS cleavable capability of eFSY, we designed a software AixUaa for identifying Uaa-induced MS cleavable cross-linked peptides, and AixUaa is also compatible with noncleavable cross-linked peptides. Finally, we applied this integrated workflow to map direct interactomes of Trx1 and SELM in live cells and revealed that SELM directly interacts with multiple regulators of calcium flux.

## Results

### Genetically encoded latent bioreactive Uaa eFSY in *E. coli*

eFSY (Fig. 1a) was synthesized by a five-step procedure (Supplementary Note 1). We engineered 5 amino acids (356Q, 391E, 393 V, 464 F and 490 M) of amino acid binding pocket of chPheRS-1[35] for incorporating eFSY (Supplementary Fig. 1a). His-tagged EGFP with a TAG codon at site 190 was co-expressed with different chPheRS-1 mutants in *E. coli* DH10B cells in the presence of eFSY. After His-tag purification, strong bands of full-length EGFP proteins were detected from samples of chPheRS-1 mutants (F464C, M490A; V393G, F464C; F464C, Q356N; F464C) on SDS-PAGE gel (Supplementary Fig. 1b). High resolution electrospray ionization time-of-flight MS analysis shows that chPheRS-1 mutant (V393G, F464C) is able to recognize eFSY, but a very small amount of tryptophan was also incorporated into EGFP (Supplementary Fig. 1c). Thus, we named this mutant as eFSYRS. To validate the incorporation of eFSY, maltose binding protein (MBP) and His-tagged Z protein with a TAG codon at site 24 was co-transformed with eFSYRS

in *E. coli* DH10B cells. Full length MBP-Z(E24eFSY) proteins were obtained with the yield of 2.24 mg/L in the presence of 1 mM eFSY in growth media (Fig. 1b–c). The peak (m/z 50835.0) corresponding to intact MBP-Z(E24eFSY) was observed by intact protein mass measurement (Fig. 1d). Glu-C digested peptides of MBP-Z(E24eFSY) were also analyzed by tandem MS, showing continuous high mass accuracy b and y ions to confirm the successful incorporation of eFSY at MBP-Z site 24 (Fig. 1e–f). eFSY incorporation on His-tagged affibody protein was also confirmed by western blot and MS analysis (Supplementary Fig. 2).

### Cross-link affibody and Z protein with eFSY

FSY is able to cross-link with His, Tyr and Lys[16]. To test cross-linking ability of eFSY, we constructed His-tagged affibody (7Ala, 7Lys, 7His, 7Tyr) mutants and introduced eFSY at site 24 of MBP-Z based on the structure of affibody-Z protein complex (Fig. 1g). His-tag purified affibody mutants were individually incubated with MBP-Z(E24eFSY) at 37 °C. Western blot analysis showed strong cross-linking bands of affibody (7 Lys, 7His, 7Tyr) with MBP-Z(E24eFSY) but not with affibody 7Ala (Fig. 1h). Biotin was successfully attached to eFSY incorporated proteins (Fig. 1i). Cross-linking sites and biotinylated peptides were confirmed by tandem mass spectra (Fig. 1j and Supplementary Fig. 3), and a cross-linked peptide TSVDNAFNHE(9)-ILHLPNLNJE(9) was significantly enriched (~10 folds) by mono-avidin beads (Fig. 1k–l). Digested samples showed much simpler mass spectra after cross-linked peptides enrichment (Supplementary Fig. 4). We believe this enrichment after cross-linking will enhance the detection of cross-linked peptides.

We also identified another biotinylated peptide MTSVDNAFNHE(10)-ILHLPNLNJE(9) with a protein N-terminal methionine residue remaining in the alpha chain (Supplementary Fig. 5a). However, its fragmentation pattern on the tandem mass spectrum is different from biotinylated TSVDNAFNHE(9)-ILHLPNLNJE(9). Limited fragment ions were found to match the alpha chain of MTSVDNAFNHE(10)-ILHLPNLNJE(9). These unassigned peaks on the tandem mass spectrum can be explained after considering the fragmentation of the cross-linking bond (Supplementary Fig. 5b–d). MTSVDNAFNHE(10)-ILHLPNLNJE(9) was also enriched by Monomeric Avidin beads (Supplementary Fig. 5e). After checking other biotinylated peptides, we found that Uaa fragment ions were observed for biotinylated peptides with J-H and J-K cross-linking, but few on J-Y cross-linking (Supplementary Fig. 6). Moreover, they were also detected for unbiotinylated U-H and U-K cross-linked peptides (Supplementary Fig. 7). Therefore, eFSY is a latent bioreactive Uaa with MS cleavage capability. We checked previously published FSY cross-linking data[9] and found that FSY induced protein cross-linking is MS cleavable as well (Supplementary Fig. 8).

### Identifying Trx1 interacting proteins in *E. coli*

Trx1 plays a cytoprotective role by reducing oxidized cysteines and the cleavage of disulfide bonds. Thus, it is sensitive to cellular redox environment[36], and it is important to identify substrates and direct interacting proteins of Trx1 in physiological environments. BprY, FSK and FSY have been applied to identify direct binding proteins of Trx1 in live cells[8,26]. To explore the sensitivity of eFSY on the detection of PPIs, we incorporated eFSY into site 62 on the protein interaction interface of Trx1 in *E. coli* (Fig. 2a). Multiple cross-linking bands were observed on western blot and these bands can be labelled with biotin (Fig. 2b–c), suggesting that Trx1 interacting proteins were successfully captured by eFSY. After His-tag purification, biotin labeling and streptavidin enrichment of cross-linked peptides, a total of 311 cross-linked peptides corresponding to 150 proteins were identified in all 3 technical replicates (Fig. 2d–e and Supplementary Data 3). This result demonstrated that eFSY can achieve highly efficient protein cross-linking.

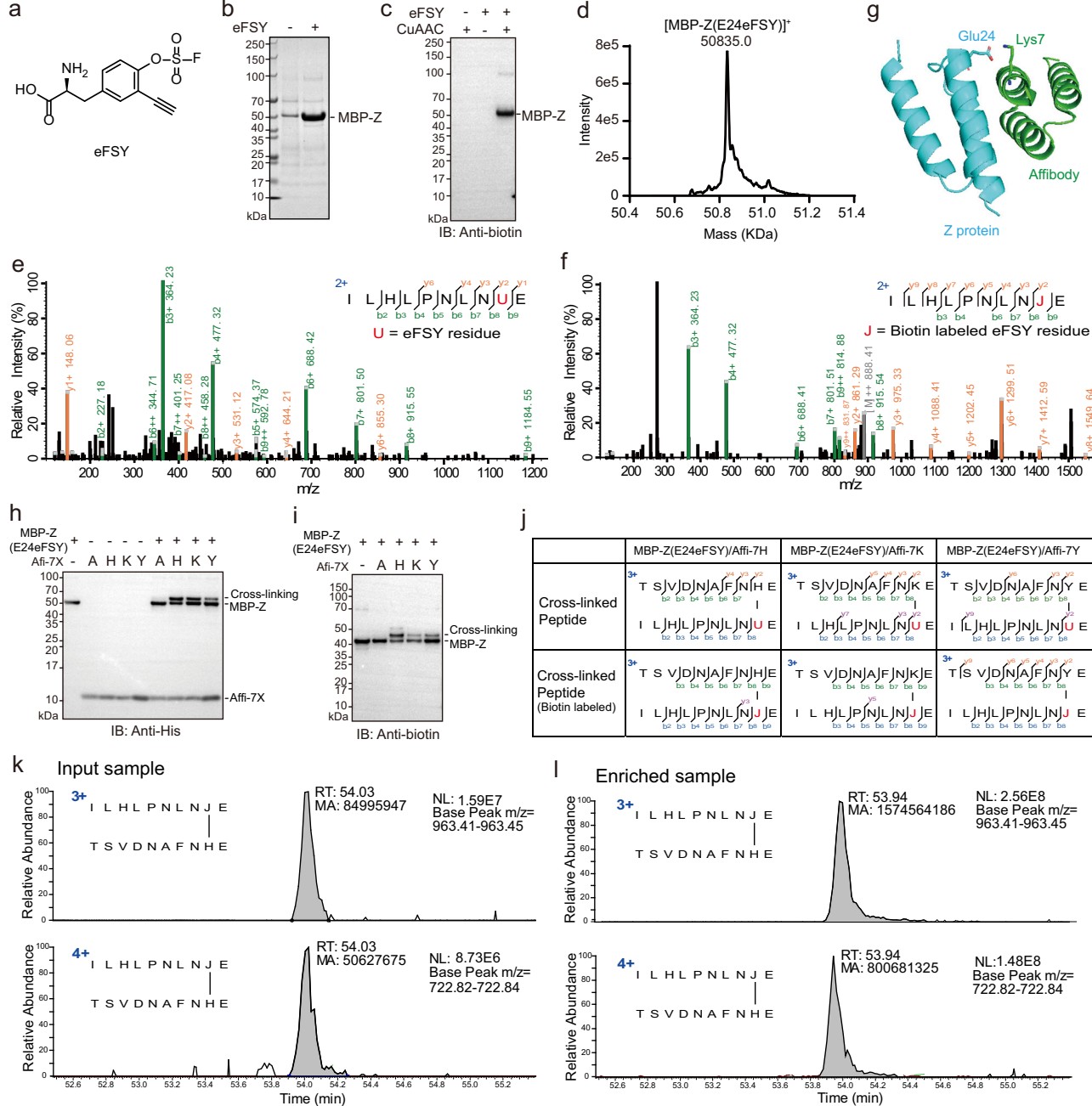

**Fig. 1 | Incorporation of eFSY into proteins in E. coli with genetic code expansion and eFSY mediated cross-linking with cross-linked peptides enrichment. a** Chemical structure of eFSY. eFSY: enrichable fluorosulfate-L-tyrosine. (**b**) SDS-PAGE gel of His-tag purified MBP-Z(E24eFSY). **c** Western blot analysis showing MBP-Z(E24eFSY) labeled by biotin via click chemistry. CuAAC: Copper-catalyzed azide-alkyne cycloaddition. **d** Intact protein mass analysis of MBP-Z(E24eFSY). **e** Tandem mass spectrum of eFSY incorporated peptide of MBP-Z(E24eFSY) ('U' represents eFSY residue in this study, but not selenocysteine). **f** Tandem mass spectrum of a biotin-labeled peptide derived from MBP-Z(E24eFSY). **g** Proximal sites, Glu24 and Lys7, on the protein structure of affibody/Z complex (PDB ID 1LP1). Glu24 was used for eFSY incorporation, and Lys7 was used

for target amino acid incorporation. (**h**) Western blot analysis showing cross-linking between MBP-Z(E24eFSY) and affibody(K7X). **i** Biotin labeling of cross-linked MBP-Z/affibody products. **j** MS/MS fragmentation patterns of cross-linked peptides and biotin-labeled peptides of MBP-Z/affibody cross-linking. **k** Extracted ion chromatograms (XICs) of cross-linked peptides from input sample (top panel: charge 3+ peptide precursor; bottom panel: charge 4+ peptide precursor). **l** XICs of cross-linked peptides from enriched sample, cross-linked peptides were effectively enriched by Monomeric Avidin beads (top panel: charge 3+ peptide precursor; bottom panel: charge 4+ peptide precursor). RT: retention time, MA: peak area. Source data are provided as a Source Data file.

## Development of a software AixUaa for identifying MS cleavable Uaa cross-linking

When we manually checked the spectra of identified cross-linked peptides, we found that one cross-linked peptide should be U·H cross-linking, but not U·Y cross-linking (Fig. 2f and g). This misassignment was caused by the fragmentation of the MS-cleavable eFSY-H cross-

linking bond (Fig. 2h and i). Therefore, the tandem MS spectrum is showing the fragmentation mixture of two linear peptides. We summarized characters of fragment ions from different cross-linking types of eFSY and found that internal fragment ions as a result of cleaving eFSY cross-linking bond showed high percentage in U·H and U·K cross-linking, but not in U·Y cross-linking (Fig. 3a). For cross-linked peptides

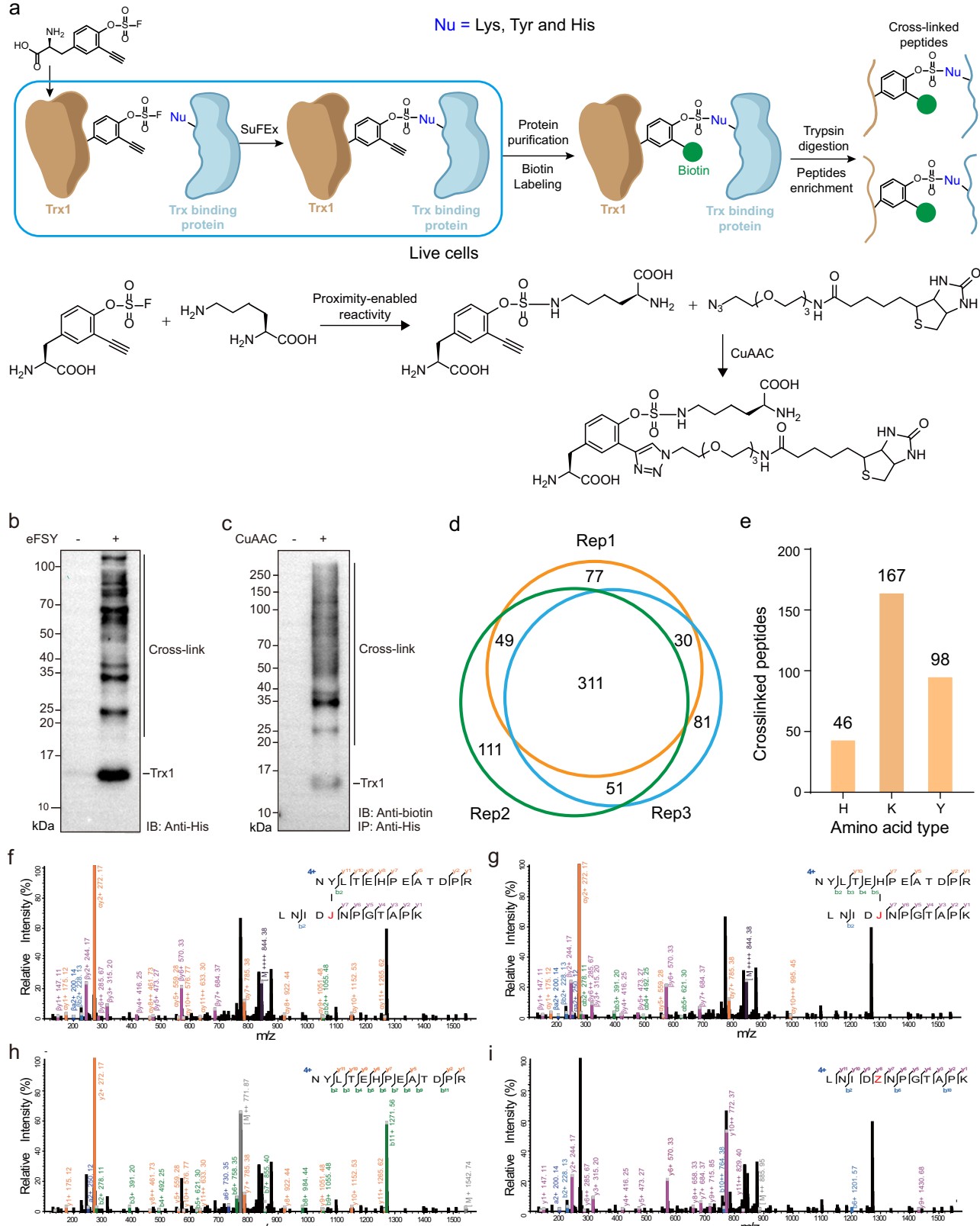

**Fig. 2 | Incorporation of eFSY into Trx1 to capture and identify direct interactome of Trx1 in E. coli. a** Experimental workflow of enrichment and identification of eFSY mediated cross-linking. Nu: nucleophilic residue. **b** Western blot analysis of *E. coli* cell lysates expressing Trx1(C35A-Q62eFSY), showing multiple endogenous proteins cross-linked to Trx1. **c** His-tag purified Trx1(C35A-Q62eFSY) cross-linked protein complexes were labeled with biotin followed by western blot analysis. **d** Cross-linked peptides identified by three replicates of MS measurements (identified with OpenUaa). **e** Types of cross-linking sites on Trx1(C35A-Q62eFSY) cross-linked endogenous proteins identified by OpenUaa. **f** Mass spectrum of cross-linked peptide with cross-linking site mis-matching. **g** Correct cross-linking site of peptide in panel f. (h) Internal fragment ions labeled on α chain of cross-linked peptide. **i** Internal fragment ions labeled on β chain of cross-linked peptide. Source data are provided as a Source Data file.

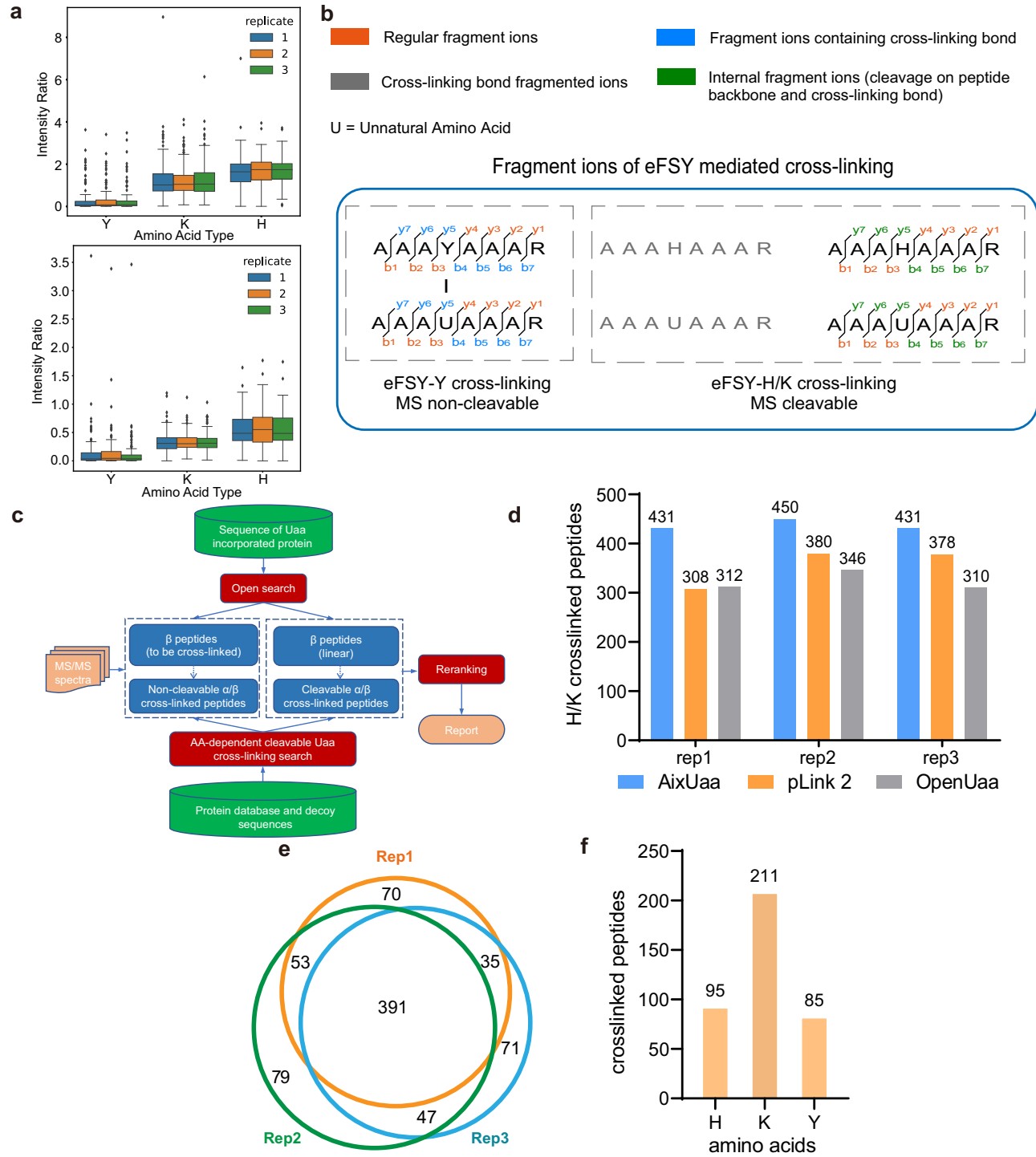

**Fig. 3 | Development of software AixUaa. a** Box plots showing the fragmentation characteristics. Top: ratio of intensity (internal fragment ions + cross-link bond fragmented ions)/intensity (regular fragment ions); Bottom: ratio of intensity (internal fragment ions)/intensity (regular fragment ions). ($n = 46$ for H, 167 for K, 98 for Y. Data are derived from 3 technical replicates as shown in Fig. 2d. In each replicate, only the best matched spectrum of each cross-linking peptide was chosen to calculate the intensity ratio). The centre line denotes the median, the box contains the 25th to 75th percentiles. The whiskers mark the minima and maxima.

Values beyond upper bounds are outliers. Source data are provided as a Source Data file. **b** Fragment ion types of eFSY-mediated cross-linking. The blue ions were not considered by traditional cleavable cross-linking identification software and the green ions were not considered by traditional non-cleavable cross-linking identification software. **c** Workflow of AixUaa. **d** eFSY-Lys/His cross-links identified by different search engine. **e** Cross-linked peptides identified by three replicates of mass spectrometry measurements with AixUaa. **f** Types of cross-linking sites on Trx1(C35A-Q62eFSY) cross-linked endogenous proteins identified by AixUaa.

identification, non-cleavable cross-linking identification software, such as pLink 2[37] and OpenUaa[9], consider fragmentation ions containing the cross-linking bond. However, few of these ions exist in the tandem mass spectra of eFSY-H cross-linking. Meanwhile, this kind of software

don't consider internal fragment ions from fragmentation of Uaa cross-linking bond, the peaks of internal fragment ions will be assigned as regular fragment ions (Fig. 3b). Therefore, non-cleavable cross-linking identification software could cause misassignments in

identification of cross-linking peptides. Cleavable cross-linking identification software depends on signature ions[27], but there are no paired signature ions in the mass spectra of eFSY-H cross-linking. eFSY induced cross-linking contained MS non-cleavable and cleavable cross-links. However, no software can perfectly decipher MS data of this kind of cross-linking, and a new searching engine is needed.

To comprehensively identify eFSY-induced protein cross-linking, we developed a searching engine AixUaa (Supplementary Note 2), which fully considers all of the fragment ions for the assignment of mass spectra and peptides (Fig. 3b). As shown in Fig. 3c, sequences and cross-linking sites of Uaa incorporated peptides (β peptides) were generated from a sequence of Uaa incorporated protein. Cross-linking partners (α peptides) were searched against protein database in MS cleavable or MS non-cleavable mode. Identified peptides were reranked with semi-supervised, machine learning-based, fine-scoring algorithm and false discovery rate (FDR) was calculated. To evaluate the capability of AixUaa to identify Uaa-mediated MS cleavable cross-linking, we have constructed 4 simulated datasets of mass spectra of U-H/K cross-linking (Supplementary Table 1): dataset Sim1 (1000 mass spectra, only 1 H/K in α chain, full of b/y ions), dataset Sim2 (1000 mass spectra, only 1 H/K in α chain, full of b/y ions on β chain, full of y ions on α chain, partial of b ions on α chain), dataset Sim3 (1000 mass spectra, only 1 H/K in α chain, full of b/y ions on β chain, full of y ions on α chain, no b ions on α chain), and dataset Sim4 (200 mass spectra, at least 1 H/K and Y in α chain, full of b/y ions). AixUaa showed high identification precision than existing database search engine on these simulated datasets (Supplementary Fig. 9). On thioredoxin pull-down samples, searching with AixUaa identified more U-H/K cross-links (Fig. 3d) and a total of 391 cross-linked peptides, corresponding to 184 proteins (Fig. 3e, Supplementary Data 3). 7 of 12 previously identified cross-linked peptides by FSY were also found in this study, and importantly, we identified 32 times more cross-linked peptides than FSK or FSY[26]. Well-known substrates of Trx1, such as Tpx, BCP, PAPR and MsrA, were identified by eFSY[38], which validated our cross-linking strategy. For example, the inter-protein cross-linking of Trx1 and PAPR was identified in all three replicates of tandem MS measurements with high confidence (Supplementary Fig. 10a and b), and cross-linking sites are located on the protein interaction interface of these two proteins (Supplementary Fig. 10c). We found that eFSY mainly targeted Lys residues (Fig. 3f), likely due to the high abundance of Lys on protein surfaces.

Search tool for the retrieval of interacting genes/proteins (STRING) network analysis reveals that Trx1 cross-linking proteins formed an organized network and had large number of PPIs (Supplementary Fig. 11a). Kyoto encyclopedia of genes and genomes enrichment showed pathways involved by Trx1 interacting proteins (Supplementary Fig. 11b). Because Trx1 is an essential regulator of cell metabolism[39,40], we identified Trx1 direct interacting proteins involved in citrate cycle and amino acid metabolism. Trx1 can also stabilize ribosomal proteins[41,42]. As expected, ribosomal proteins were significantly enriched in our Trx1 cross-linking protein dataset. Gene ontology term analysis showed that Trx1 interacting proteins involved in biological processes of translation, metabolism and oxidoreduction (Supplementary Fig. 11c).

## Incorporation of eFSY in mammalian cells

We next tested eFSY incorporation and its cross-linking capability in mammalian cells. HEK293T cells were co-transfected with EGFP-151TAG and pNEU-eFSYRS plasmids in the presence of eFSY. The amber stop codon of the EGFP 151 site was suppressed by eFSYRS to express full length of EGFP. Successful expression of eFSY incorporated EGFP was confirmed by fluorescence microscopy (Fig. 4a). The incorporation efficiency is quite robust even in lower eFSY concentrations (Supplementary Fig. 12). We co-expressed EGFP containing a TAG codon at site 190 and pNEU-eFSYRS plasmids in the presence of

eFSY, and confirmed successful incorporation of eFSY at EGFP 190 site by intact protein MS measurement and tandem MS analysis (Fig. 4b–c). To determine the cross-linking capability of eFSY in mammalian cells, we incorporated eFSY at site 103 within the dimer interface of a dimeric protein glutathione S-transferase (GST) in HEK293T cells. eFSY mediated dimeric cross-linking band was successfully detected by western-blot (Fig. 4d). Further tandem MS analysis identified three inter-chain cross-linked peptides. Cross-linked amino acid residues showed close proximity to the eFSY incorporation site (Fig. 4e). These cross-linking sites were precisely mapped by high-quality tandem mass spectra (Fig. 4f–h). These results suggest that eFSY can be efficiently incorporated in mammalian cells, and it has high cross-linking efficiency in live cells.

## Identifying direct interactome of Selenoproteins in mammalian cells

The trace element selenium presents as selenocysteine (Sec) at the active site of selenoproteins, and 25 selenoproteins have been identified in mammals. Most of selenoproteins are involved in the regulation of redox homeostasis in the cells. However, the functions of selenoproteins are not well characterized, due to the difficulty of obtaining native forms of selenoproteins in preparative quantities as selenocysteine is encoded by stop codon TGA[43]. We applied eFSY to mapping direct interactome of selenoproteins, with a goal to better understand the functions of selenoproteins by analyzing their direct interacting proteins. First, we incorporated eFSY into selenocysteine sites of selenoproteins (SELW, SELM, MSRB1, SEP15 and GPX1) in live cells. As a result, many cross-linking bands were observed (Supplementary Fig. 13), suggesting that eFSY can capture interacting proteins of selenoproteins in live cells.

SELM is one of the least studied mammalian selenoproteins. It has a CXXSec motif and potential oxidoreductase activity. Recently, a protective effect of SELM on non-alcoholic fatty liver disease, cytotoxic effect of recombinant SELM on human glioblastoma cells, and SELM's promotion of leptin signaling and thioredoxin activity have been reported[44–46]. To map direct interacting proteins of SELM, we incorporated eFSY into site 48 of human SELM (Fig. 5a). After expressing SELM (Sec48eFSY) in HEK293T cells, western blot analysis showed that SELM was cross-linked with its endogenous binding proteins and these cross-linked protein complexes can be labeled with biotin via click chemistry (Fig. 5b–c). SELM (Sec48eFSY) and its cross-linked partners were purified by Strep IP, followed by biotin labeling at protein-level and digestion by trypsin. Cross-linked peptides were enriched by Monomeric Avidin beads and subjected to MS analysis. A total of 24 cross-linked peptides were identified by all three technical replicates (Fig. 5d and Supplementary Data 3), corresponding to 10 direct interacting proteins of SELM. SELM is located to perinuclear structures corresponding to Endoplamic Reticulum (ER) and Golgi. As expected, the identified interacting proteins of SELM are also mainly located in ER and Golgi (Supplementary Fig. 14a), which validated the specificity of our approach. The interacting proteins of SELM are involved in isomeration and oxidoreduction of disulfide bond and calcium ion binding (Supplementary Fig. 14b). Through STRING network analysis, we found that direct partners of SELM formed a subnetwork (Supplementary Fig. 14c). We randomly chose four proteins HSPA5, PRDX4, PLGT3 and Calreticulin and carried out Co-IP experiments to validate their interactions with SELM. Western blot analyses demonstrated that they indeed interacted with SELM (Fig. 5e).

We also identified 2 inter-protein cross-linked peptide between PRDX4 and SELM (Supplementary Fig. 15) at cross-linking sites PRDX4(K250) and PRDX4(K265), which are close to PRDX4(C124) and PRDX4(C245) on protein structure (Fig. 5f). This result suggests that SELM may catalyze oxidoreduction of PRDX4(C124) and PRDX4(C245). We further applied disulfide trapping approach[47] to validate the interaction between PRDX4 and SELM. PRDX4 forms homodimer with inter-

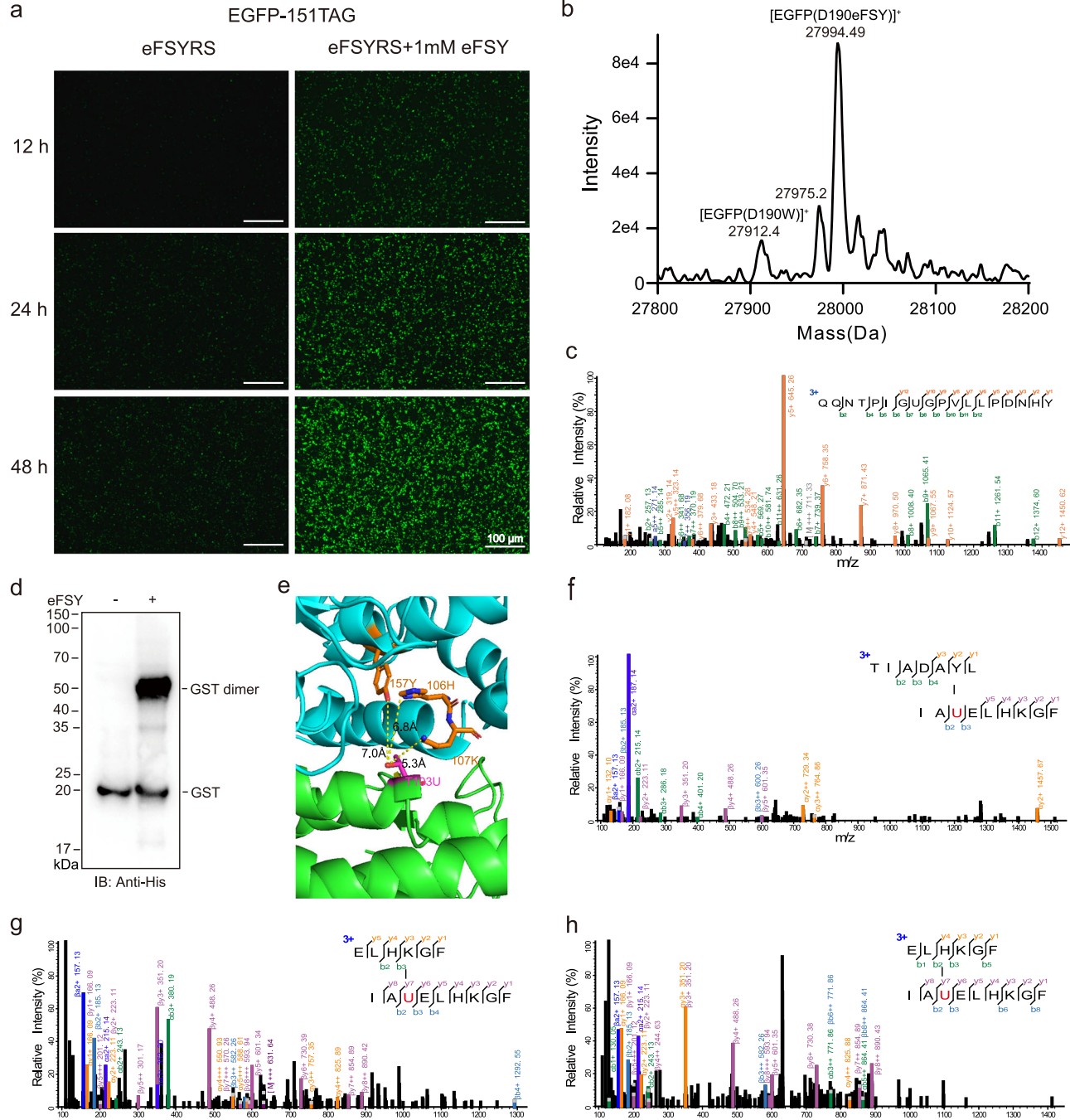

**Fig. 4 | Genetically encoding eFSY into proteins in mammalian cells.**
**a** Fluorescence images of cells expressing EGFP(Y151eFSY), showing successful incorporation of eFSY into EGFP in HEK293T cells. **b** Intact protein mass analysis of EGFP(D190eFSY) expressed in HEK293T cells. **c** Tandem mass spectrum of eFSY incorporated peptide derived from EGFP(D190eFSY). **d** Western blot analysis of HEK293T cell lysates expressing GST(T103U), showing cross-linking of GST dimer

in live cells. Source data are provided as a Source Data file. **e** Identified inter-chain cross-links mapped onto the protein structure of GST (PDB ID 1AOF). The eFSY incorporation site was shown in magenta sticks. The cross-linking sites were shown in orange sticks. **f–h** Tandem mass spectra of inter-chain cross-linked peptides derived from GST.

chain disulfide bond (C124-C245), and 5 homodimers are connected by inter-chain disulfide bond (C51-C51) to form a ring-like decamer[48]. We constructed SELM(Sec48S), SELM(C45S-Sec48C) and mutants (M1-6) of PRDX4 for disulfide trapping experiments (Fig. 5g). Covalent complexes of SELM(C45S-Sec48C)/PRDX4 M5 and SELM(C45S-Sec48C)/PRDX4 M6 were observed (Fig. 5h), indicating that SELM(C45S-Sec48C) can trap PRDX4 and there is an interaction between SELM and PRDX4. SELM has thioredoxin-fold domain and redox-active motif (CXXSec), suggesting that it may has oxidoreductase function[49].

## Discussion

In comparison with eFSY-Tyr cross-linking, sulfamate of eFSY-His and eFSY-Lys cross-linking can be protonated in gas phase, which will result in the cleavage of cross-linking bond during High-energy Collision Dissociation. His and Lys are basic amino acids and have high proton affinity than Tyr, this will stabilize fragment ions after the cleavage of cross-linking bonds, and thus, increase the abundance of these fragment ions. Cleavage of cross-linking bond will generate two linear peptide ions, and these linear peptide ions will be further fragmented

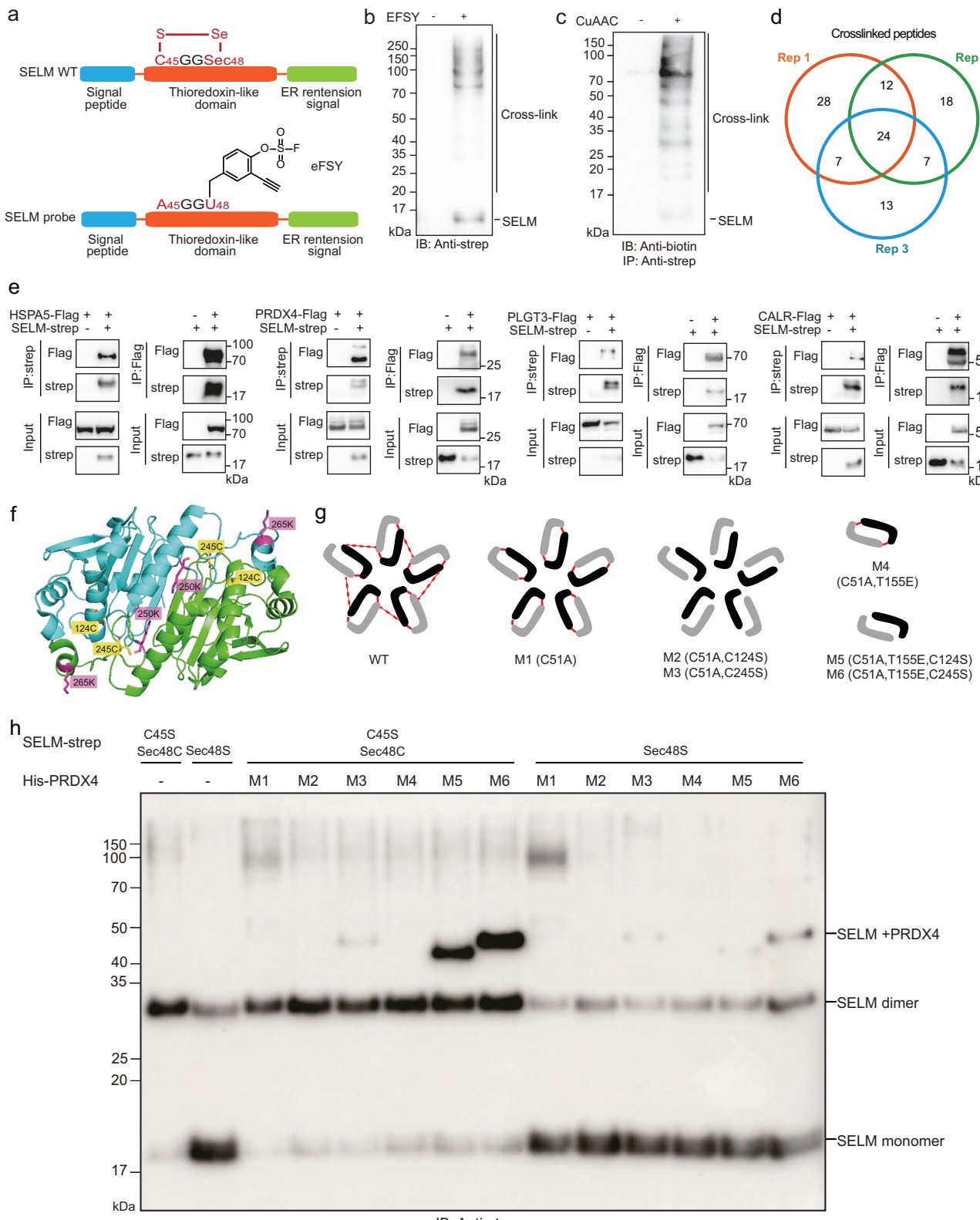

**Fig. 5 | Identifying direct interactome of SELM via eFSY induced live cell cross-linking. a** The design of the SELM probe in which eFSY was incorporated at site 48 and Cys45 was mutated to Ala ('Sec' represents selenocysteine). **b** Western blot analysis of HEK293T cell lysates expressing SELM(C45A-Sec48U). **c** Strep-tag pur-ified protein complexes cross-linked by the SELM probe were labeled with biotin, followed by western blot analysis. **d** Inter-protein cross-linked peptides identified by MS. **e** Validation of PPI between SELM and binding proteins (HSPA5, PRDX4,

PLGT3 and CALR). **f** SELM-PRDX4 cross-linking sites mapped onto the protein structure (PDB ID 3TJF). The cross-linking sites were shown in magenta sticks. Cysteines of PRDX4 were shown in yellow sticks. **g** Mutants (M1-6) of PRDX4 (picture adapted from PMID: 21057456). **h** Disulfide trapping experiment showing SELM cross-linked with PRDX4 by inter-protein disulfide bond. Source data are provided as a Source Data file.

to generate internal fragment ions. Linear peptide ions which contain eFSY residue are not stable, as the sulfate group is prone to be cleaved (Supplementary Fig. 16).

Sulfur-fluoride exchange reaction has been widely used for developing covalent drugs[17,18,20,50], screening ligandable tyrosines and lysines[51], and capturing protein interactors[22,26]. A key step of above mentioned studies is site-specifically identifying cross-linking sites. If database search engine doesn't consider MS cleavage on sulfamate, it could cause mis-assignment of cross-linking site (Fig. 2 g–i). Therefore, our study is helpful for the design of new peptide identification strategy to precisely map covalent drug binding sites or identify new covalent druggable proteins. At the same time, high sensitivity of eFSY can be useful in identifying off-targets of covalent drugs and studying mechanisms of drug resistance. In the future, eFSY can be extended to study protein-DNA and protein-RNA interactions. In addition, the reactivity of eFSY can be fine-tuned for different biological applications.

SELM is highly expressed in brain[52], and it has been suggested to have neuroprotective functions[53]. Calcium signaling is related to pathogenesis of neurodegenerative diseases[54], and SELM has been shown to decrease calcium responses of cells under oxidative stress and reduce cell apoptosis[53]. However, how SELM regulate calcium signaling is not clear. Here, we identified direct binding proteins of SELM: Calumenin, endoplasmic reticulum chaperone BiP (BIP), Reticulocalbin-1 and Calreticulin. All of these interactors are involved in the regulation of $Ca^{2+}$ flux. Sarcoplasmic/endoplasmic reticulum $Ca^{2+}$-ATPase (SERCA) is responsible for reuptake cytosolic $Ca^{2+}$ from cytosol to ER, Calumenin interacts with loop4 of SERCA2 and inhibits activity of SERCA2 to regulate $Ca^{2+}$-cycling in cardiomyocytes[55]. On the one hand, the binding of BIP to ER luminal loop 7 of Sec61α limits $Ca^{2+}$ leakage from ER[56]. On the other hand, BIP interacts with IP3R1 and IP3R1 alters the $Ca^{2+}$ signal by sensing ER stress through BIP[57]. Reticulocalbin-1 also interacts with IP3R1 to inhibit calcium release[58]. Calreticulin recruits oxidoreductase ERp57 to the luminal Loop 4 of SERCA 2b, and ERp57 promotes disulfide bond formation in L4 under high concentration of calcium. This inhibits pump activity of SERCA 2b[59]. Our data suggest that SELM may be involved in the regulation of calcium flux through its direct binding proteins.

Genetically encoded latent bioreactive Uaas have been used to capture transient protein-protein interaction in live cells. However, in comparison with small molecular cross-linkers, less cross-linking warheads are introduced into proteins, and less cross-links were generated between two proteins by latent bioreactive Uaas. If the Uaa cross-linked peptides are too long, too short, too hydrophobic or too hydrophilic, they could cause difficulties in the general MS identification. We believe it could be beneficial to combine genetically encoded cross-linking and small molecule cross-linking for identification of protein interactions. Firstly, latent bioreactive Uaas can be used to fix transient protein interactions, which will stabilize protein complexes for downstream purification and analysis. Secondly, small molecule cross-linker can be applied to add more cross-links between fixed proteins to facilitate mass spectrometry identification. In the meantime, we need to develop high performance software for double cross-linking identification because products of double cross-linking are complex (Supplementary Fig. 18).

In summary, we chemically synthesized a latent bioreactive, enrichable and MS cleavable Uaa, eFSY, and obtained highly efficient aminoacyl-tRNA synthetase which can recognize eFSY in both *E. coli* and mammalian cells. eFSY successfully captured transient and direct protein-protein interactions site specifically. With peptide-level enrichment, high-throughput identification of PPIs in *E. coli* and mammalian cells can be achieved. Based on MS cleavable capability of eFSY cross-linking, we also developed software AixUaa to increase the sensitivity of MS identification and sequence coverage of cross-linked peptides.

## Methods

### Reagents
Unless otherwise noted, all commercial reagents were used without further purification. Primers were synthesised by Tsingke Biotech and sequences are provided in Supplementary Data 1. Company names and catalog numbers of all commercial reagents are provided in Supplementary Data 2. The following antibodies were used: anti-His tag (Mouse, Proteintech, cat. no. HRP-66005, 1:10000), anti-FLAG tag (Mouse, Proteintech, cat. no. HRP-66008, 1:10000), anti-Strep-tag II (Mouse, Wuhan Dian Biotechnology, cat. no. 2098, 1:10000), anti-biotin (HRP-linked Antibody, Cell Signaling Technology, cat. no. 7075 S, 1:2500), Goat anti-Mouse IgG (H + L) Secondary Antibody (Jackson ImmunoResearch, Code. 115-035-003, 1:10000).

### Software
Thermo Scientific Xcalibur 4.5.445.18 was used to collect the LC-MS/MS data. SCIEX OS 3.1.6.44 used to collect the LC-MS data. Tanon 5200 was used to collect the immunoblots. ImageView 4.11 was used to detect fluorescent images. pLink 2.3.11, OpenUaa and AixUaa were used for identification of cross-linking peptides. pFind 3.2.0 was used for identification of regular peptides. Graphpad Prism 8.0.2 was used for statistical analysis. PyMOL 2.3.3 was used to visualize of crystal structures of proteins. ChemDraw 19.0 was used to draw structures of molecular. Cytoscape 3.10.0 was used to visualize the interaction networks.

### Plasmid construction
To incorporate eFSY in *E. coli*, the pBK-eFSYRS plasmid was constructed by introducing double mutations (V393G and F464C) into the chimeric phenylalanyl-tRNA synthetase[35]. The genes encoding target proteins including MBP-Z, affibody and Trx1 were cloned into pNEG vector bearing the chimeric tRNA 3C11[60]. The Amber stop codon TAG was introduced at the incorporation site by site-directed mutagenesis. The gene encoding affibody was also cloned into pBAD vector, and the plasmids of Afb4A-7X (X= His, Lys, Tyr, Ala) were constructed by introducing mutations at site 4 and site 7.

To incorporate eFSY in mammalian cells, eFSYRS and four copies of tRNA 3C11 were cloned into pNEU vector. The genes encoding target proteins including His-tagged EGFP, His-tagged GSTA and selenoproteins (SELW, SELM, MSRB1, SEP15, GPX1) with Strep-tag II were cloned into pRK5M vector. The Amber stop codon TAG or other mutations were introduced by site-directed mutagenesis. The gene encoding PRDX4 was cloned into pET22b vector with a N terminal His tag. The N terminal signal peptide containing amino acids 1-37 was removed. Mutants of PRDX4 were obtained by site-directed mutagenesis. Genes encoding HSPA5, PRDX4, PLGT3 and CALR were cloned into pRK5M vectors with C-terminal Flag tag for Co-IP experiments.

### Protein expression
The plasmids pNEG-MBP-Z-24TAG, pNEG-affibody-39TAG or pNEG-Trx1(C35A-Q62TAG) were co-transformed with pBK-eFSYRS into DH10B competent cells, separately. pBAD-Afb4A-7X was directly transformed into DH10B competent cells and pET22b-PRDX4 were transformed into BL21-DE3 competent cells. The transformed cells were plated on LB agar plate with 50 μg/mL ampicillin and 50 μg/mL kanamycin (ampicillin only for Afb4A-7X and PRDX4) and incubated overnight at 37 °C. A single colony was picked and incubated with 1 mL LB medium, which was further diluted into 50 mL LB (25 mL LB for Afb4A-7X and PRDX4). When OD600 reached 0.8 (0.6 for Afb4A-7X and PRDX4), the cell culture was separated into two tubes and induced with 0.2% arabinose in the absence or presence of 1 mM eFSY (without eFSY for Afb4A-7X and PRDX4 was induced by 1 mM IPTG for 4 h at 37 °C). After incubated at 30 °C for 6 h (Trx1-C35A-Q62eFSY for 12 h), cell pellets were collected by centrifugation at 4000 g for 30 min at 4 °C and stored at −80 °C.

## Protein purification

For the purification of His tagged MBP-Z(E24eFSY), affibody(D39eFSY-D40N), Afb4A-7X, Trx1(C35A-Q62eFSY), EGFP(D190eFSY) and GSTA(-T103eFSY), cell pellets were resuspended in 5 mL lysis buffer (50 mM Tris-HCl pH 8.0, 500 mM NaCl, 20 mM imidazole, 1% v/v Tween 20 and protease inhibitors). The cell suspension was lysed by sonication (35% output, 5 min, 1 sec off, 1 sec on) followed by centrifugation (16,200 g, 30 min, 4 °C). The supernatant was incubated with 150 μL pre-equilibrated Ni-NTA Beads at 4 °C for 1 h. Then the beads were washed with three volumes of wash buffer 1 (50 mM Tris pH8.0, 500 mM NaCl, 20 mM imidazole) and wash buffer 2 (50 mM Tris pH 8.0, 500 mM NaCl, 40 mM imidazole). The proteins were eluted with elution buffer (50 mM Tris pH8.0, 500 mM NaCl, 250 mM imidazole), and the buffer was exchanged to storage buffer (50 mM HEPES, pH 7.5, 150 mM NaCl).

For the purification of SELM(C45A-Sec48eFSY) with Strep-tag II, cell pellets were resuspended in 5 mL lysis buffer (50 mM Tris-HCl pH 8.0, 150 mM NaCl, 1 mM EDTA, 1% NP-40 and protease inhibitors). The cell suspension was lysed by sonication (20% output, 2 min, 1 sec off, 1 sec on) followed by centrifugation (16,200 g, 30 min, 4 °C). The supernatant was incubated with 500 μL pre-equilibrated Streptactin Beads 4FF at 4 °C for 2 h. Then the beads were washed with six volumes of wash buffer (50 mM Tris-HCl pH 8.0, 150 mM NaCl, 1 mM EDTA) and eluted with elution buffer (50 mM Tris-HCl pH 8.0, 150 mM NaCl, 1 mM EDTA, 20 mM D-biotin), followed by buffer exchanging to storage buffer (50 mM HEPES, pH 7.5, 150 mM NaCl).

## Incorporation of eFSY in mammalian cell

HEK293T (ATCC, CRL-3216) cells were cultured in DMEM medium supplemented with 10% fetal bovine serum and 1% penicillin−streptomycin. pNEU-eFSYRS and pRK5M-EGFP(Y151TAG) were co-transfected at 80% cell confluency. Six hours post transfection, the media containing transfection complex were replaced with fresh DMEM media with 10% FBS in the presence or absence of 1 mM eFSY. After incubation at 37 °C for 12 h, 24 h and 48 h, transfected cells were imaged. For optimization of eFSY concentration, cells were cultured under different concentrations of eFSY for 48 hours for imaging.

## Mass analysis of intact proteins

Purified proteins were analyzed on an SCIEX Triple TOF 6600 MS System equipped with an electrospray ionization (ESI) source in conjunction with SCIEX Analyst TF software. Separation and desalting were carried out on a PHENOMENEX AERIS WIDEPORE C4 Column (200 Å, 2.1×50 mm, 3.6 μm). Mobile phase A was 0.1 % formic acid in water and mobile phase B was acetonitrile. A constant flow rate of 0.3 mL/min was used. Data was analyzed using SCIEX OS-Q software. Mass spectral deconvolution was performed using SCIEX OS-Q software (version 2.0, SCIEX Corporation).

## In vitro cross-linking

A 40 μL reaction mixture containing MBP-Z-24eFSY (50 μg/mL) and Afb4A-7X (50 μg/mL) in HEPES buffer (pH 7.5) was incubated at 37 °C for 12 h. 10 ul reaction mixture was analyzed by western blot (Anti-6×His). 10 μL solution was prepared for MS analysis by in solution digestion, and the remaining 20 μL product was further labeled with biotin by click chemistry.

## Click chemistry reaction

Proteins were labeled by the addition of 1 mM Biotin-PEG3-Azide, 150 μM CuSO4, 300 μM BTTAA and 5 mM sodium ascorbate. The mixture was incubated at 30 °C for 1 hour and quenched by 5 mM EDTA. To validate the incorporation of eFSY into MBP-Z(E24TAG), purified MBP-Z(E24eFSY) was labeled with biotin and analyzed by western blot (Anti-biotin) and the remaining MBP-Z(E24eFSY) was in-solution digested for MS analysis. To validate the cross-linking of MBP-

Z(E24eFSY) and Afb4A-7X, the 20 μL solution of cross-linking product was labeled with biotin. Then half of reaction mixture was analyzed by western blot (Anti-biotin). The other half was prepared for MS analysis by in solution digestion. For enrichment of cross-linked peptides of Trx1(C35A-Q62eFSY) and SELM(C45A-Sec48eFSY), 200 μg purified protein was labeled. 2 μg was analyzed by western blot (Anti-biotin) and the remaining protein was digested for peptide enrichment.

## Protein digestion

Protein samples were precipitated with six volumes of prechilled acetone and incubated at −20 °C for 30 minutes. The precipitated proteins were centrifuged at 16,200 g for 30 minutes. The supernatant was removed. For digestion of MBP-Z(E24eFSY), affibody(D39eFSY-D40N) or cross-linking products of MBP-Z(E24eFSY) and Afb4A-7X, the pellets were resuspended with 50 mM NH4HCO3 pH 8.0. Then 0.25 μg Glu-C was added and incubated at 37 °C overnight. After reduction with 5 mM TCEP (tris(2-carboxyethyl)phosphine) for 20 min and alkylation with 10 mM iodoacetamide for 15 min in the dark, digestion was terminated by adding 5% FA (formic acid). Digested peptides were further desalted with StageTips.

For digestion of EGFP(D190eFSY) and GST(T103eFSY), the pellets were resuspended with 8 M urea in 100 mM Tris pH 8.5. After reduction and alkylation, samples were diluted to 1 M urea with 100 mM Tris pH 8.5, 10 mM CaCl2, and digested with 0.5 μg chymotrypsin at 25 °C overnight. Digestion was terminated by adding 0.5% trifluoroacetic acid, and digested peptides were further desalted with StageTips.

## Enrichment of cross-linked peptides

The biotin labeled proteins were transferred to 10 kDa ultrafiltration unit and concentrated to 100 μL. A total of 300 μL 8 M urea in 100 mM Tris pH 8.5 was added, and the solution was concentrated to 100 μL. This process was repeated four times to denature the proteins and remove the excess Biotin-PEG3-Azide. After reduction and alkylation, 300 μL 50 mM NH4HCO3 pH 8.0 was added and the solution was concentrated to 100 μL. This process was repeated four times to exchange the buffer for trypsin digestion. Trypsin was added at a 1:50 (w/w) ratio and incubated at 37 °C overnight. The digested peptides were collected and incubated with 30 μL PBS prewashed Pierce Monomeric Avidin Agarose at room temperature for 2 hours. After centrifugation and removal of supernatant, the agarose was washed with 500 μL wash buffer 1 (20 mM HEPES, 1 M KCl, pH 8.0), 500 μL PBS and 500 μL 10% ACN (acetonitrile). Every wash step was repeated three times to fully remove the non-specifically bound peptides. The bound biotinylated peptides were eluted with 200 μL elution buffer (50% ACN, 5% FA) for three times. The eluate was dried by centrifugation under a vacuum and then resuspended in 5% FA for desalting.

To enrich cross-linked peptides of MBP-Z(E24eFSY) and Afb4A-7H, a mixture of 40 μg MBP-Z(E24eFSY) and 40 μg Afb4A-7H were incubated overnight. After click chemistry and Glu-C digestion, one-tenth of the mixture was taken out as input, and the remaining peptide was enriched as described above. Finally, 0.5 μg peptide of each sample was used for LC-MS/MS analysis.

## MS analysis

The desalted peptides were dried by centrifugation under vacuum followed by resuspension in 0.1% FA for LC-MS/MS analysis. Peptide mixture was analyzed using an Easy-nLC 1200 system coupled with an Orbitrap Exploris 480 (ThermoFisher) mass spectrometer. The sample was loaded directly onto a home-made capillary column (75 μm × 20 cm, 1.9 μm C18, 5 μm tip). Mobile phase A consisted of 0.1% FA, 2% ACN and 98% H2O and mobile phase B consisted of 0.1% FA, 20% H2O and 80% ACN. A 60 min gradient (mobile phase B: 4% at 0 min, 5% at 1 min, 25% at 41 min, 37% at 54 min, 90% at 57 min, 90% at 60 min) was used at a static flow rate of 450 nl/min. The data were acquired in a data-dependent (top-30) mode. For MS1, the scan range was set to

350 – 1500 m/z at a resolution of 60,000. The AGC target was set as 1e6 with a maximum injection time of 20 ms. For MS2, the resolution was set to 15,000 with a fixed first mass of 125 m/z. The AGC target was set to 1e5 and the maximum injection time was set to 22 ms. Dynamic exclusion was set to 30 s with a 10 ppm mass tolerance around the precursor. Ions with charge state 1, 6-8 and more than 8 were excluded. Particularly, ions with charge state 3-6 were included for analysis of enriched cross-linked peptides of Trx1(C35A-Q62eFSY) and SELM(C45A-Sec48eFSY). Raw files of Trx1 and SELM and GST were collected as three technical repeats. Other LC-MS/MS experiments were performed once.

## MS data analysis

To detect the eFSY incorporation, raw files of MBP-Z(E24eFSY) and EGFP(D190eFSY) were searched against the corresponding proteins by pFind3[61]. For analysis of cross-linked peptides, raw files of cross-linked MBP-Z(E24eFSY) and Afb4A-7X were searched by AixUaa. Raw files of enriched cross-linked peptides of Trx1 were searched against the *E. coli* proteome downloaded from the UniProt database by OpenUaa, AixUaa and pLink2. Carbamidomethyl (C) was considered as fixed modification and Oxidation (M) was considered as variable modification. Search results of OpenUaa, AixUAA and pLink2 were filtered with the false discovery rate (FDR) of 5% at the peptide-spectrum match (PSM) level.

Raw files of enriched cross-linked peptides of SELM were first searched against the human proteome downloaded from the UniProt database by pFind3. Then the identified proteins of three technical replicates were combined as the database for AixUaa analysis. Search results of AixUaa were filtered with the false discovery rate (FDR) of 5% at the peptide-spectrum match (PSM) level. For all workflows, Carbamidomethyl (C) was considered as fixed modification and Oxidation (M) was considered as variable modification.

## Statistics and Reproducibility

The experiments in the Figs. 1b, c, h, i, 2b, c, 4a, d, 5b, c, e and h were repeated twice with similar results.

## Reporting summary

Further information on research design is available in the Nature Portfolio Reporting Summary linked to this article.

## Data availability

The mass spectrometry data generated in this study have been deposited in the ProteomeXchange Consortium under accession code PXD047084. Source data of box blots in Fig. 3a and uncropped versions of gels or blots are provided with this paper. Any Supplementary Information (methods, figures, notes), and Supplementary Data (reagents, DNA sequences and list of cross-linking peptides) are available in this paper. Source data are provided with this paper.

## Code availability

AixUaa is developed in Python and is freely available. The latest software version is available at https://github.com/BUAA-LiuLab/AixUaa.

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

## Acknowledgements

We thank technical assistance from LSI core facility, Life Sciences Institute (LSI), Zhejiang University and GIBH analytical center, Guangzhou Institutes of Biomedicine and Health (GIBH), Chinese Academy of Sciences. This work was supported by the outstanding youth fund of Zhejiang Province (LR20B050001 to B.Y.), the National Key R&D Program of China (2022YFF0608402 to B.Y., 2021YFA1301602 to C.L., 2021YFA1301603 to C.L. and 2019YFA0904500 to S.T.), the Chinese National Natural Science Funds (22374128 to B.Y., 22074132 to B.Y., 91953103 to B.Y. and 32171442 to C.L.), the special COVID-19 program of the Sino-German Center for Research Promotion (C-0023 to B.Y.), Basic Research Project of Guangzhou Institutes of Biomedicine and Health, Chinese Academy of Sciences (GIBHBRP23-02 to S.T.), Open Project Program of the State Key Laboratory of Proteomics (SKLPO201806 to B.Y.) and the Fundamental Research Funds for the Central Universities.

## Author contributions

D.D.L., D.W., J.T.C. and Q.W. contributed equally to this work. D.D.L. conducted the majority of the experiments and data analysis. D.D.L. and D.W. screened eFSYRS. J.T.C., Q.W., and C.L. developed the software AixUaa. Y.L. and J.T. chemical synthesized eFSY. T.Y.Z. performed mass spectrometry experiments. L.Z. provided critical suggestions on this

project. K.S., P.L. and F.Y. performed bioinformatics analysis. S.T. designed the eFSY. B.Y. wrote the manuscript. S.T. and B.Y. supervised the project.

## Competing interests

The authors declare the following competing interests: B.Y., S.T., D.D.L., Zhejiang University and Guangzhou Institutes of Biomedicine and Health have filed a patent application (Chinese patent application number: 2024105555109) on the design and application of the fluorosulfate containing tyrosine analog eFSY. All other authors declare no competing interests.
