## [Peer Review File · Nature Communications]

Reviewers' Comments:

Reviewer #1:

Remarks to the Author:

The authors describe the synthesis, incorporation, and application of a latent bioreactive unnatural amino acid eFSY, which features an additional enrichable alkyne group for click labeling of biotin. They also discovered that the linkage between eFSY and Lys or His was MS cleavable. To cope with such cleavage, the authors further advanced the search engine to account for the related MS ions, developing a new software called AixUaa for the identification of cross-linked peptides. With all these advancements, they demonstrated enhanced detection sensitivity of cross-linked peptides using the model protein Trx in *E. coli* cells. They further applied the eFSY-based crosslinking to SELM, a selenoprotein, in mammalian cells to identify its direct interacting proteins. Overall, the development of eFSY and AixUaa software would significantly improve our ability in identifying the direct interactome of proteins in cells with enhanced sensitivity and accuracy. This technology will complement and has potential to be superior to the existing photo-cross-linking approaches. I support its publication after the following points are addressed appropriately.

Page 3, Abstract: 1st sentence 'Proximity-based chemical cross-linkable unnatural amino acids (Uaas) have been widely used in the development of covalent drugs and...'.

These Uaas have been termed "Latent bioreactive unnatural amino acids". This term should be used instead of coming up with a new term.

Page 4, Introduction: 'Genetically encoded cross-linkable unnatural amino acids (Uaas) enable capturing protein interactions in live cells, 5,6'.

This review, <https://doi.org/10.1016/j.cbpa.2023.102285>, should be cited here.

Page 4, Introduction: 'In addition, FSY has been applied to developing covalent protein drugs, nanobody, 16,17,18'.

This reference, DOI: 10.1021/acscentsci.3c00288, should be cited here.

Reference 21, describing covalent protein drugs for neutralizing SARS-CoV-2, should also be included here.

Page 4-5, Introduction: 'However, it is unknown whether FSY incorporated protein binders have off-target effect,'

The off-target effect has been investigated in this reference, DOI: 10.1039/D3SC01921G, through other methods. This reference should be cited here.

Page 6, Results, Genetically encoded chemical cross-linkable Uaa eFSY in *E. coli*.

Figure 1b indicates that in the absence of eFSY some full-length protein was also made; Figure S1c also indicates Trp was mis-incorporated in the presence of eFSY. Does eFSYRS misincorporate Trp? If so, it should be stated in the main text. The current writing assumes that eFSYRS incorporates eFSY only.

Page 6, 'FSY is able to cross-link with His, Tyr and Lys.' Reference 15 should be cited after this sentence to support this statement.

Figure 1k-l: It is unclear what are the differences in the top panel and bottom panel – clarify in figure legend.

Page 7, paragraph starting with 'It is interesting that we also identified another biotinylated peptide MTSVDNAFNHE(10)ILHLPNLNJE(9) (Supplementary Fig. 5a).'

The writing in this paragraph is confusing.

Page 8, 'However, few of these ions exist in the tandem mass spectra of eFSY-H cross-linking, which can cause misassignments in identification of cross-linking peptides.'

This would reduce signal/noise and detect less cross-linked peptides, but how can this cause misassignment if the assignment holds high standards? Please clarify.

Figure 3b is also unclear to nonexperts. A more detailed figure legend is needed.

Page 11, 'The incorporation is quite robust with high incorporation efficiency in lower eFSY concentrations (Supplementary Fig. 12).'

A better way to substantiate this statement is to quantify fluorescence intensity via flow cytometry.

Figure 4b, any misincorporation of Trp in mammalian cells?

Page 12, Discussion, 'Sulfur-fluoride exchange (SuFEx) reaction has been widely used for developing covalent drugs, 16,17,44'

These two references should be cited here: DOI: 10.1021/acscentsci.3c00288, and DOI: 10.1021/acscentsci.3c00288

Page 12, Discussion, 'capturing protein interactors. 19'

Reference 23 should be included here.

Reviewer #2:

Remarks to the Author:

The manuscript described the development of an enrichable proximity-based chemical cross-linkable unnatural amino acids eFSY, based on a previously developed FSY (Fluorosulfate-L-tyrosine). They discovered that eFSY and FSY are MS-cleavable, and took advantage of this property in improving the crosslink identification with a new software program AixUaa specifically developed for it. The manuscript used a well-selected example 'Selenoprotein' to study the PPIs near their selenocysteine sites. The potential applications of this technique are very attractive, and the paper should be published after addressing the following issues.

1. To let the readers figure out if the method fits for their specific problem. The authors should add some descriptions in introduction or discussion about the disadvantages of using cross-linkable Uaas over the chemical small molecular crosslinkers. The limitation of this type of technique.

Uaas are expensive regarding synthesis and a lot of material is needed in a single experiment. For a general protein, one may need to test a lot of insertions of the Uaa at different positions to capture all potential PPI.

2. Just to indicate the potential of eFSY, in the introduction, you can add the description that a different version of FSY (SFY) capturing protein-glycan interaction after 'and capturing protein-RNA interactions.'

3. Any reference for this sentence? 'Lastly, because the ionization efficiency of noncleavable cross-linked peptides is low and it is not easy to obtain large fragment ions across the crosslinking site, relative low fragment ions coverage of cross-linked peptides is typically observed.'

4. To facilitate the usage of the technique, please release the full vector sequence of pNEU-eFSYRS (or related) in materials or deposit them to addgene or somewhere else if possible. Please add description of 'chPheRS' or give references in their first appearance.

5. In page 7 (merged pdb), the "cross-linked peptide TSVDNAFNHE(9)-ILHLPNLNJE(9) was significantly enriched (~15-20 folds) by mono-avidin beads (Fig. 1k-l)." Did this 15-20 folds number simply be calculated from intensity comparison? It seems like if normalize the total intensity based on Fig.S4, the folds will be ~10 folds for this special example.

6. In page 7, 'It is interesting that we also identified another biotinylated peptide MTSVDNAFNHE(10)-ILHLPNLNJE(9) (Supplementary Fig. 5a).' Please tell me why it is interesting? Loss of 'M'? Sorry if I miss something.

7. In page 8. 'When we manually checked spectra of identified cross-linked peptides, we found that one crosslinked peptide should be U-H cross-linking, but not U-Y cross-linking (Fig. 2f and g).' Is it possible that both U-H and U-Y exist and they coelute?

8. In page 8, 'Cleavable cross-linking identification software (CXIS) depends on signature ions,²⁴ but there are no paired signature ions in mass spectra of eFSY-H cross-linking.' There is another software program named MetaMorpheusXL (<https://doi.org/10.1021/acs.jproteome.8b00141>), which can identify cleavable crosslinked peptides and does not rely on identifying signature ions. It seems like it will work.

Regarding the identification, as we always already know the half peptide (the linker peptide with Uaa or 'beta' peptides) of the Uaa crosslinked peptides, a simple way is to add it as a labile modification (with fragments as signature ions) and search with the traditional programs such as MSFragger-Labile (<https://doi.org/10.1016/j.mcpro.2023.100538>, seems like it will work). Please explain.

9. Later the data analysis only compare AixUaa with pLink and OpenUaa, which is biased if the other methods work. I believe AixUaa are optimized to be one of the best for this specific problem, still please add new comparisons if possible.

10. Minor, in method MS data analysis, the parameters of pLink2 are not described.

Reviewer #3:

Remarks to the Author:

In this manuscript entitled "Characterize direct protein interactions with cross-linkable, enrichable and mass spectrometry cleavable unnatural amino acids in live cells," the authors investigated the usability of unique cross-linkable unnatural amino acids (Uaas). The concept of the present study is interesting; however, several concerns were included. My comments were as follows:

Comments:

1. The quality of the abstract was poor. The authors should summarize the novel findings of this study precisely. The description of the background was too much.
2. In the introduction, the authors described that "Genetically encoded cross-linkable unnatural amino acids (Uaas) enable capturing protein interactions in live cells, 5, 6 and protein direct interactions and their interaction interfaces can be mapped by the identification of Uaa mediated cross-linked peptides. 7, 8"; however, this sentence was uncertain. Please explain these issues with specific examples.
3. In Fig 1e and 1j, the authors used the term "U" to indicate eFSY residue. This description was very complicated because "U" indicates selenocysteine.
4. Overall, the abbreviation of this manuscript was too much and complicated.
5. Finally, the authors identified the interaction between SELM and PRDX4. Please confirm this interaction by using other previous methods and should demonstrate the usability of the novel method.

Response to reviewer #1 (Black: reviewer comments, Blue: our response)

Reviewer #1 (Remarks to the Author):

The authors describe the synthesis, incorporation, and application of a latent bioreactive unnatural amino acid eFSY, which features an additional enrichable alkyne group for click labeling of biotin. They also discovered that the linkage between eFSY and Lys or His was MS cleavable. To cope with such cleavage, the authors further advanced the search engine to account for the related MS ions, developing a new software called AixUaa for the identification of cross-linked peptides. With all these advancements, they demonstrated enhanced detection sensitivity of cross-linked peptides using the model protein Trx in *E. coli* cells. They further applied the eFSY-based crosslinking to SELM, a selenoprotein, in mammalian cells to identify its direct interacting proteins. Overall, the development of eFSY and AixUaa software would significantly improve our ability in identifying the direct interactome of proteins in cells with enhanced sensitivity and accuracy. This technology will complement and has potential to be superior to the existing photo-cross-linking approaches. I support its publication after the following points are addressed appropriately.

Page 3, Abstract: 1st sentence ‘Proximity-based chemical cross-linkable unnatural amino acids (Uaas) have been widely used in the development of covalent drugs and...’ .

These Uaas have been termed “Latent bioreactive unnatural amino acids” . This term should be used instead of coming up with a new term.

Response: We thank the reviewer for pointing this out. We have corrected it in the manuscript.

Page 4, Introduction: ‘Genetically encoded cross-linkable unnatural amino acids (Uaas) enable capturing protein interactions in live cells, 5,6’ .

This review, <https://doi.org/10.1016/j.cbpa.2023.102285>, should be cited here.

Response: We thank the reviewer for this suggestion. We have cited this review in the manuscript.

Page 4, Introduction: ‘In addition, FSY has been applied to developing covalent protein drugs, nanobody, 16,17,18’ .

This reference, DOI: 10.1021/acscentsci.3c00288, should be cited here.

Reference 21, describing covalent protein drugs for neutralizing SARS-CoV-2, should also be included here.

Response: We thank the reviewer for these suggestions. We have cited this paper in the manuscript

Page 4-5, Introduction: ‘However, it is unknown whether FSY incorporated protein binders have off-target effect,’

The off-target effect has been investigated in this reference, DOI: 10.1039/D3SC01921G, through other methods. This reference should be cited here.

Response: We thank the reviewer for this suggestion. We have cited the paper and corrected the sentence in manuscript.

Page 6, Results, Genetically encoded chemical cross-linkable Uaa eFSY in E. coli.

Figure 1b indicates that in the absence of eFSY some full-length protein was also made; Figure S1c also indicates Trp was mis-incorporated in the presence of eFSY. Does eFSYRS misincorporate Trp? If so, it should be stated in the main text. The current writing assumes that eFSYRS incorporates eFSY only.

Response: We agree with the reviewer. We have added this sentence ‘but a very small amount of tryptophan was also incorporated into EGFP’ to the main text.

Page 6, ‘FSY is able to cross-link with His, Tyr and Lys.’ Reference 15 should be cited after this sentence to support this statement.

Response: We thank the reviewer for this suggestion. We have cited this paper in the manuscript.

Figure 1k-l: It is unclear what are the differences in the top panel and bottom panel – clarify in figure

legend.

Response: We apologize for causing confusion. We have clarified it in figure legend.

Page 7, paragraph starting with ‘It is interesting that we also identified another biotinylated peptide MTSVDNAFNHE(10)ILHLPNLNJE(9) (Supplementary Fig. 5a).’

The writing in this paragraph is confusing.

Response: We apologize for causing confusion. We have optimized writing of this paragraph.

Page 8, ‘However, few of these ions exist in the tandem mass spectra of eFSY-H cross-linking, which can cause misassignments in identification of cross-linking peptides.’

This would reduce signal/noise and detect less cross-linked peptides, but how can this cause misassignment if the assignment holds high standards? Please clarify.

Response: We apologize for the confusion about this sentence. We have re-written this part in the manuscript.

Figure 3b is also unclear to nonexperts. A more detailed figure legend is needed.

Response: We apologize for the confusion. We have redrawn Fig. 3b and re-written figure legend of Fig. 3b.

Page 11, ‘The incorporation is quite robust with high incorporation efficiency in lower eFSY concentrations (Supplementary Fig. 12).’

A better way to substantiate this statement is to quantify fluorescence intensity via flow cytometry.

Response: We thank the reviewer for this suggestion. We have performed flow cytometry analysis on HEK293T cells with treatments of different concentration of eFSY. The results were updated to Supplementary Fig. 12b.

Figure 4b, any misincorporation of Trp in mammalian cells?

Response: Yes, there is misincorporation of Trp in mammalian cells. We have labelled the related peak

on Fig. 4b.

Page 12, Discussion, ‘Sulfur-fluoride exchange (SuFEx) reaction has been widely used for developing covalent drugs, 16,17,44’

These two references should be cited here: DOI: 10.1021/acscentsci.3c00288, and DOI: 10.1021/acscentsci.3c00288

Response: We thank the reviewer for this suggestion. We have cited this paper in the manuscript.

Page 12, Discussion, ‘capturing protein interactors. 19’ .

Reference 23 should be included here.

Response: We thank the reviewer for this suggestion. We have cited this paper in the manuscript.

Reviewer #1 (Remarks on code availability):

I don't have appropriate expertise to review the code.

Response to reviewer #2

Reviewer #2 (Remarks to the Author):

The manuscript described the development of an enrichable proximity-based chemical cross-linkable unnatural amino acids eFSY, based on a previously developed FSY (Fluorosulfate-L-tyrosine). They discovered that eFSY and FSY are MS-cleavable, and took advantage of this property in improving the crosslink identification with a new software program AixUaa specifically developed for it. The manuscript used a well-selected example ‘Selenoprotein’ to study the PPIs near their selenocysteine sites. The potential applications of this technique are very attractive, and the paper should be published after addressing the following issues.

1. To let the readers figure out if the method fits for their specific problem. The authors should add some descriptions in introduction or discussion about the disadvantages of using cross-linkable Uaas over the chemical small molecular crosslinkers. The limitation of this type of technique.

Uaas are expensive regarding synthesis and a lot of material is needed in a single experiment. For a general protein, one may need to test a lot of insertions of the Uaa at different positions to capture all potential PPI.

Response: We thank the reviewer for his/her comments. We have discussed the disadvantages of using cross-linkable Uaas in the section of discussion. We are developing small molecule covalent labeling strategy for selection of Uaa incorporation sites, this strategy will make selection of Uaa incorporation sites easier. We will publish it soon.

2. Just to indicate the potential of eFSY, in the introduction, you can add the description that a different version of FSY (SFY) capturing protein-glycan interaction after ‘ and capturing protein-RNA interactions.’

Response: We thank the reviewer for his/her comments. We cannot add this potential application of eFSY to the introduction, because cross-linking products of eFSY and glycan is prone to be hydrolyzed (Yang B. et al, JACS, 2019).

3. Any reference for this sentence? ‘Lastly, because the ionization efficiency of noncleavable cross-linked peptides is low and it is not easy to obtain large fragment ions across the crosslinking site, relative low fragment ions coverage of cross-linked peptides is typically observed.’

Response: We thank the reviewer for his/her comments. We have cited 2 papers to this sentence in the manuscript (Barysz, HM. et al., Molecular & Cellular Proteomics, 2018; Kolbowski L. et al., Analytical Chemistry, 2022).

4. To facilitate the usage of the technique, please release the full vector sequence of pNEU-eFSYRS (or related) in materials or deposit them to addgene or somewhere else if possible. Please add description

of ‘chPheRS’ or give references in their first appearance.

Response: We thank the reviewer for his/her comments. We have added sequences of tRNA-synthetase and tRNA to supplementary methods.

5. In page 7 (merged pdb), the” cross-linked peptide TSVDNAFNHE(9)-ILHLPLNJE(9) was significantly enriched (~15-20 folds) by mono-avidin beads (Fig. 1k-l).” Did this 15-20 folds number simply be calculated from intensity comparison? It seems like if normalize the total intensity based on Fig.S4, the folds will be ~10 folds for this special example.

Response: We thank the reviewer for his/her comments. We have corrected it in the manuscript.

6. In page 7, ‘It is interesting that we also identified another biotinylated peptide MTSVDNAFNHE(10)-ILHLPLNJE(9) (Supplementary Fig. 5a).’ Please tell me why it is interesting? Loss of ‘M’ ? Sorry if I miss something.

Response: We apologize for the misleading of this sentence ‘It is interesting that’, we have deleted it.

7. In page 8. ‘When we manually checked spectra of identified cross-linked peptides, we found that one crosslinked peptide should be U-H cross-linking, but not U-Y cross-linking (Fig. 2f and g).’ Is it possible that both U-H and U-Y exist and they coelute?

Response: There is a possibility that both U-H and U-Y exist. However, there are not enough fragment ions to support the existing of U-Y cross-linking.

8. In page 8, ‘Cleavable cross-linking identification software (CXIS) depends on signature ions,²⁴ but there are no paired signature ions in mass spectra of eFSY-H cross-linking.’ There is another software program named MetaMorpheusXL (<https://doi.org/10.1021/acs.jproteome.8b00141>), which can identify cleavable crosslinked peptides and does not rely on identifying signature ions. It seems like it will work.

Regarding the identification, as we always already know the half peptide (the liner peptide with Uaa

or ‘beta’ peptides) of the Uaa crosslinked peptides, a simple way is to add it as a labile modification (with fragments as signature ions) and search with the traditional programs such as MSFragger-Labile (<https://doi.org/10.1016/j.mcpro.2023.100538>, seems like it will work). Please explain.

Response: We thank the reviewer for his/her comments. We have tried MetaMorpheusXL, it supports identification of U-H and U-K cross-linking, but not U-Y cross-linking. MSFragger-Labile can be used to identify EFSY mediated cross-links, but fragment ions of beta chain were only considered as ‘diagnostic ions’ for mass spectra filtering and they were not considered in peptide scoring. It is different from labile modification, actually there are multiple fragment ions on beta chain (Fig. R1). If these fragment ions were not considered in peptide scoring, it will lower sensitivity of cross-linked peptides identification.

Figure R1. Fragment ions on Uaa incorporated peptide. (a) Tandem mass spectrum of crosslinked peptide CrI(K15)-Trx1(Q62eFSY). (b) Multiple fragment ions on cross-linked peptide beta chain.

9. Later the data analysis only compare AixUaa with pLink and OpenUaa, which is biased if the other methods work. I believe AixUaa are optimized to be one of the best for this specific problem, still please add new comparisons if possible.

Response: We haven't compared AixUaa with MetaMorpheusXL, because it doesn't support identification of U-Y cross-linking. We have added new comparison with MSFragger-Labile (Fig. R2).

Figure R2. Comparison of identified cross-linking peptides with AixUaa, pLink2, OpenUaa and MSFragger-Labile. (a) U-H/K cross-linked peptides identified by different database search engine (left), mean of identified cross-linked peptides (right). (b) U-H/K/Y cross-linked peptides identified by different database search engine (left), mean of identified cross-linked peptides (right).

10. Minor, in method MS data analysis, the parameters of pLink2 are not described.

Response: We thank the reviewer for this suggestion. We have corrected it.

Reviewer #2 (Remarks on code availability):

Code is not available, as the paper didn't mention the software program is open source.

The software is easy to run.

Response: We thank the reviewer for his/her comments, we have uploaded code to github.

Reviewer #3 (Remarks to the Author):

In this manuscript entitled “Characterize direct protein interactions with cross-linkable, enrichable and mass spectrometry cleavable unnatural amino acids in live cells,” the authors investigated the usability of unique cross-linkable unnatural amino acids (Uaas). The concept of the present study is interesting; however, several concerns were included. My comments were as follows:

Comments:

1. The quality of the abstract was poor. The authors should summarize the novel findings of this study precisely. The description of the background was too much.

Response: We thank the reviewer for this suggestion. We have removed some backgrounds from the abstract.

2. In the introduction, the authors described that “Genetically encoded cross-linkable unnatural amino acids (Uaas) enable capturing protein interactions in live cells,5, 6 and protein direct interactions and their interaction interfaces can be mapped by the identification of Uaa mediated cross-linked peptides.7,8” ; however, this sentence was uncertain. Please explain these issues with specific examples.

Response: We thank the reviewer for this suggestion. For example, 1. Previously, latent bioreactive Uaa has been applied to study direct interaction of Urocortin-I/CRF1R and map the protein interaction interface (Coin I. et al., Cell, 2013); 2. In this study, we have identified well-known binding proteins of

Trx1, such as PAPR. Cross-linking peptide Trx1(Q62EFSY)-PAPR(191Y) was identified, and the cross-linking sites locate on protein interaction interface (Fig. R3).

Figure R3. Cross-linking sites locate on protein interaction interface of Trx1/PAPR.

3. In Fig 1e and 1j, the authors used the term “U” to indicate eFSY residue. This description was very complicated because “U” indicates selenocysteine.

Response: We apologize for the confusion about the term ‘U’. We have clarified it in figure legend of Fig. 1e. We used ‘Sec’ to represent selenocysteine in Fig. 5a.

4. Overall, the abbreviation of this manuscript was too much and complicated.

Response: We thank the reviewer for his/her comments. We have removed following abbreviations: AP-MS, NXIS, CXIS, KEGG, HCD, SuFEx, CALU, CALR and RCN1.

5. Finally, the authors identified the interaction between SELM and PRDX4. Please confirm this interaction by using other previous methods and should demonstrate the usability of the novel method.

Response: We thank the reviewer for his/her comments. We have validated SELM/PRDX4 interaction with Co-IP and disulfide-trapping approach. To demonstrate the usability of eFSY enrichment, we have performed SELM(Sec48FSY) SELM(C45A-Sec48FSY) IP and 3 cross-linked peptides were identified. Peptide level enrichment with eFSY is useful.

Figure R4. Identification of direct interacting proteins of SELM with FSY. (a) Western blot of SELM(Sec48FSY) and SELM(C45A-Sec48FSY) Strep-IP, showing multiple endogenous proteins cross-linked to SELM. (b) List of proteins cross-linked to SELM(C45A-Sec48FSY) in mammalian cells obtained by cross-linked peptides identification.

Reviewers' Comments:

Reviewer #2:

Remarks to the Author:

The authors addressed all my comments. I support the publication of this paper.

Reviewer #3:

Remarks to the Author:

The authors completely answered my queries, and I would like to recommend the publication of this manuscript.